# Acceleration of Skin Wound-Healing Reactions by Autologous Micrograft Tissue Suspension

**DOI:** 10.3390/medicina56070321

**Published:** 2020-06-29

**Authors:** Shiro Jimi, Satoshi Takagi, Francesco De Francesco, Motoyasu Miyazaki, Arman Saparov

**Affiliations:** 1Central Lab for Pathology and Morphology, Faculty of Medicine, Fukuoka University, Fukuoka 814-0180, Japan; 2Departments of Plastic, Reconstructive and Aesthetic Surgery, Faculty of Medicine, Fukuoka University, Fukuoka 814-0180, Japan; stakagi@fukuoka-u.ac.jp; 3Department of Reconstructive and Hand Surgery, Azienda Ospedaliero Universitaria “Ospedali Riuniti” di Ancona, 60126 Ancona, Italy; fran.defr@libero.it; 4Department of Pharmacy, Fukuoka University Chikushi Hospital, Fukuoka 818-8502, Japan; motoyasu@fukuoka-u.ac.jp; 5Department of Medicine, School of Medicine, Nazarbayev University, Nur-Sultan 010000, Kazakhstan; asaparov@nu.edu.kz

**Keywords:** micrograft, wound-healing, epidermis, granulation, neovascularization, histopathology

## Abstract

*Background and objectives*: Skin grafting is a method usually used in reconstructive surgery to accelerate skin regeneration. This method results frequently in unexpected scar formations. We previously showed that cutaneous wound-healing in normal mice is accelerated by a micrograft (MG) technique. Presently, clinical trials have been performed utilizing this technology; however, the driving mechanisms behind the beneficial effects of this approach remain unclear. In the present study, we focused on five major tissue reactions in wound-healing, namely, regeneration, migration, granulation, neovascularization and contraction. *Methods:* Morphometrical analysis was performed using tissue samples from the dorsal wounds of mice. Granulation tissue formation, neovascularization and epithelial healing were examined. *Results:* The wound area correlated well with granulation sizes and neovascularization densities in the granulation tissue. Vascular distribution analysis in the granulation tissue indicated that neovessels extended and reached the subepidermal area in the MG group but was only halfway developed in the control group. Moreover, epithelialization with regeneration and migration was augmented by MG. Myofibroblast is a known machinery for wound contraction that uses α-smooth muscle actin filaments. Their distribution in the granulation tissue was primarily found beneath the regenerated epithelium and was significantly progressed in the MG group. *Conclusions:* These findings indicated that MG accelerated a series of wound-healing reactions and could be useful for treating intractable wounds in clinical situations.

## 1. Introduction

Delay in wound-healing not only prolongs hospital stay and increases medical cost, but also decreases the quality of life of patients. Wound-healing is a fundamental tissue reaction, by which wounds heal after granulation tissue formation and tissue regeneration at the sites of tissue defect. If such reactions do not progress appropriately, especially in patients with metabolic disorders including diabetes, nutritional deficiency and blood flow disturbance, treatment becomes challenging [1,2]. Recent wound treatments, such as cultured human dermis [3] and artificial dermis [4], as well as negative-pressure wound preparation [5], demonstrate a great capability of accelerating wound-healing. Basic fibroblast growth factor (bFGF) is also widely adopted for various kinds of wound treatments because it directly stimulates the growth of granulation tissue including fibroblasts and endothelial cells [6]. The harmful impact of chronic wounds is significant worldwide. From a global perspective, the annual medical cost for patients with chronic wounds is set to increase up to 3.5 billion dollars in 2021 [7]. However, the ease of obtaining reliable technology at a reasonable cost for the treatment of the world’s chronically wounded patients remains a challenge. For example, in Japan, the medical cost for patients with severe bedsores is currently over 15 billion yen/year [8], which has become a serious socioeconomic problem.

The wound-healing process is composed of inflammation, granulation tissue formation, wound contraction and epidermal regeneration and migration [9]. These processes must occur sequentially and interactively, but independently. Granulation tissue plays an important role in not only filling defected parts, but also acting as a scaffold for vascular networks and newly formed epidermis [10]. Granulation tissue is composed of fibroblasts and extracellular matrix including collagens. In the matrix, neovascularization takes place by angiogenesis and vasculogenesis [11]. These events finally terminate after full epidermal coverage of the wounded surface. Such tissue events may be perturbed by aggravating factors derived from the deteriorated disease conditions, resulting in chronic and intractable wounds.

The micrograft (MG) technique, which adopts an autologous tissue grafting method, was established by Reverdin in 1869 [12]. It initially started with hair grafting with small pieces of minced scalp with hair roots. Recently, a modified method named the Rigenera protocol [13,14], was reported, in which a subject’s own tissues are cut into microtissues of approximately ≤50 µm in size using a rotating blade. The graft may contain resident cells, tissue stem and progenitor cells, as well as various tissue factors including cytokines and growth factors. This renewal method has been used for different pathologic lesions, such as chronic ulcers [2,15,16], exaggerated scars [17] and wound opening [18]. Riccio et al. (2019) [19] recently performed a multicenter clinical analysis using a MG technique in complicated traumatic ulcers with large loss of skin and soft tissue in the lower limbs of patients with domestic or work related injuries. The aim of reconstructive treatment is the morphologic and functional restoration. The autologous MG technique utilizes a new concept in regenerative medicine for severe traumatic wounds. The powerful clinical results, integrated with the experimental research, establish that this procedure is able to stimulate skin regeneration through a real regenerative process. In our previous study, we showed that this technique is effective for wound-healing in a humanized wound model in mice [20]. However, its mechanisms of action on wounded tissue are still unclear. In this investigation, as a continuation of our MG study, we focused on the five aforementioned major tissue repair reactions that lead to wound-healing (Figure 1) to explore their major histologic factors in rodents using MG technique. However, it is well accepted that the wound-healing process in human skin is different from that in an animal. Regenerative healing primarily occurs with epidermal regeneration in humans, however, in animals, especially in mice, wound contraction is the major response. To overcome this problem, we have established an animal model with a splinted wound [20]. Using this model, we morphometrically examined the effects of MG technique.

## 2. Materials and Methods

### 2.1. Animal Experiments

This animal study was approved by the Fukuoka University Animal Experiment Committee (No. 1507848:3 July 2015) and was in compliance with the institution’s animal care guideline. At total of 30 male C57BL/6 mice aged 6–10 weeks (Japan SLC, Inc., Shizuoka, Japan) were used and all procedures were conducted under aseptic conditions. Mice were anesthetized with atmosphere exposure of isoflurane (Wako Pure Chemical Industries, Ltd., Osaka, Japan) or intraperitoneal injection of pentobarbital (somnopentyl; KYORITSU SEIYAKU, Tokyo, Japan). At the end of the study, the mice were sacrificed by pentobarbital injection and arterial hemorrhage and wound tissue was obtained.

To accurately estimate epidermal regenerative healing, a previously established splint model using a splint under the dermis was utilized [21]. In brief, mice were anesthetized and dorsal hair was removed using a commercial depilatory. A circular mark (1 cm in diameter) was made at the center of the lumbar area. The marked skin was excised with scissors, and a doughnut-shaped plastic splint (outer diameter of 28 mm; inner-hole diameter of 18 mm) was inserted beneath the skin near the wound defect, and then the splint was fixed to the skin with surgical silk thread (six stitches). Thereafter, the splinted wound was covered with a polyurethane film dressing (Tegaderm, SUMITOMO 3 M, Tokyo, Japan). To prevent thread removal by the mice, they were dressed and fixed with a silicon-tight vest to avoid gnawing off the threads by keeping their heads from turning.

### 2.2. MG Technique on Wound-Healing

Preparation of the skin tissue suspension solution for MG was reported in our previous manuscript [21]. In brief, after removing the dorsal hair, skin tissue (1 cm in diameter) was totally excised with scissors. The epidermal layer of the skin was then scraped away, which was histologically confirmed, and the tissue was washed two times with saline solution. The tissue was dissected into small pieces (approximately 1 × 1 mm). Tissue pieces were minced with a rotating blade, namely Rigeneracons^®^ (Human Brain Wave Srl., Via Pinerolo, Torino, Italy), and passed through the blade holes (50 μm in diameter), by which tissue suspension was obtained. The tissue suspension (100 μL) in a 96-well multi plate was used for measurement of absorbance determination (450 nm/550 nm) using a microplate reader (iMark, Bio-Rad Laboratories, Inc., Hercules, CA, USA). According to our previous result for optimal density of tissue suspension [20], tissue suspension in 100 μL/well was initially adjusted to OD = 1.0, and diluted two times with saline, which was used for MG. On day 3 after wounding, the wound surface was exposed to 70% ethanol for 1 min to kill superficial cells on the wounded tissue, mimicking a human chronic wound with superficial necrosis. The tissue solution or saline solution (200 µL) was inoculated on the wound surface (MG group and control group, respectively).

### 2.3. Wound Tissue Evaluation

Macroscopic evaluation: Wound images with scale indication were obtained from directly above the tissue samples using a digital camera (NEX-C3, Sony, Tokyo, Japan). The wound area percentage (WA%) against initial area was measured using a computer-assisted morphometric analyzer (VH Analyzer, VH-H1A5, KEYENCE Co., Osaka, Japan).

Microscopic evaluation: Dorsal skin tissue was dissected from the sacrificed mice. This tissue, together with the splint, was fixed in 10% buffered formaldehyde (pH 7.4) for 2 days. Two cross-cut tissue samples from each wound (approximately 5 mm thick) were excised. Paraffin blocks were prepared by using a tissue processor (Tissue-Tek VIP Premier, SAKURA, Nagano, Japan); following which, 4-μm-thick tissue sections were cut with a microtome (RM2235, Leica Biosystems, Nußloch, Germany).

### 2.4. Histological Examination

The paraffin section was stained with hematoxylin and eosin and Masson’s trichrome. To detect neovessels and myofibroblasts, rabbit anti-mouse CD31 antibody (30-times dilution: Dianova GmbH, Hamburg, Germany) and rabbit anti-mouse α-smooth muscle cell actin (α-SMA) antibody (3000-times dilution: Abcam plc, Tokyo, Japan) were used, respectively. After treatment with an enhancing reagent (EnVision Kit, DAKO Japan, Inc., Tokyo, Japan), immunohistologic localization of α-SMA was visualized by 3,3’-diaminobenzidine. Hematoxylin was used for counterstaining.

### 2.5. Morphometrical Analysis

WA%: Macroscopic wound evaluation was performed during the study using a computer-assisted program (VH-analyzer). To obtain an accurate evaluation of the wound area regression, the WA% at the time of measurement was calculated as follows: WA% = wound area on day 13/wound area on day 3 × 100.

Total epithelial length (TEL) from wound edge, defined as the length of epithelial growth from the dermal cutting edge after wounding, was morphometrically measured using a computer-assisted program (VH-analyzer). Two epithelial lengths were measured: contractive epithelial length (CEL) and regenerative epithelial length (REL). The TEL was the sum of CEL and REL. The contracted epidermis was characterized by normal dermal structures with hair follicles and a dermal–muscular coat, whereas the regenerative epidermis grown on granulation tissue did not have these characteristics.

Granulation tissue area: The granulation tissue that formed beneath the adjacent non-epithelialized zone was evaluated using a computer-assisted program (VH-analyzer). A zonal granulation tissue on picture (880 µm in surficial length, using 10× objective lens) was selected (Figure 2A); the granulation tissue area was morphometrically measured.

Neovascularization: Using the same image of the granulation tissue stained with CD-31 antibody, a transparent sheet with horizontal lines at equal intervals (18.5 µm) was placed over the previously acquired image. Using digitizing images, the total numbers of CD-31 positive stains across the lines in the granulation tissue was calculated as the total neovascular density. The spot density on each line was also calculated, which represented the distribution of neovascularization at different depths of granulation tissue.

Myofibroblast distribution: myofibroblasts with α-SMA expression was determined with histochemical staining. After taking images, the stained area was highlighted using Photoshop Elements (Ver. 11: Adobe Systems, Inc., San Jose, CA, USA), and its positive area/granulation area was calculated as a percent area of α-SMA expression in granulation using a computer-assisted program (VH-analyzer). For whole tissue detection of α-SMA expression on wounded tissue with/without MG treatment, wound tissues were resected and fixed in 5% formalin. After washing in phosphate-buffered saline (pH 7.4), all muscles and fats were removed, and specimens were incubated overnight in FITC-labeled anti-α-SMA antibody (50-times-dilution: Sigma-Aldrich, MO, USA) at 4 °C. After incubation, tissue specimens were incubated in 4’,6-diamidino-2-phenylindole. After washing, tissue specimens were incubated in a tissue clearing reagent (FocusClear FC-101, CelExplorer Co., Ltd., Hsinchu, Taiwan) for 1 h. The tissue sample was mounted by aqueous mounting media (Perma Flour, Thermo Science, Waltham, MA, USA) and observed under a fluorescent microscope (BZ-X800, Keyence Co., Osaka, Japan).

### 2.6. Statistical Analyses

Values were expressed as the mean ± standard deviation. Two values were compared using Student’s *t*-test. Spearman’s rank-correlation coefficient was used to assess the associations between WA%, granulation tissue area, the total numbers of CD-31 positive spots (neovessels points), REL, CEL, TEL and α-SMA expression area (%) each to others. All statistical analyses were performed using JMP^®^ 14 (SAS Institute, Inc., Cary, NC, USA). A *p*-value of 0.05 was considered statistically significant.

## 3. Results

### 3.1. Histological Alterations and Their Relations

Wound area, granulation, neovascularization, α-SMA distribution and epithelial healing including REL, CEL and TEL on day 13 were quantified (Table 1). All values except for neovessels and REL showed significant correlations.

### 3.2. Granulation Tissue

WA% greatly decreased in the MG group than in the saline control group (Table 2). Using transversal sections, the area of tissue granulation near the fronts of the regenerative epidermis were measured. Both were more than two times greater in the MG group (Table 2). The relationship between the WA% and granulation tissue area and thickness were negatively correlated (Table 1).

### 3.3. Neovascularization

The representative pictures for CD31-positive neovascularization in granulation tissue on days 13 in the saline control and MG groups are shown in Figure 2A. The extent of neovascularization in the granulation tissue was morphometrically analyzed. Total neovascular density was significantly greater (more than 3-times) in the MG group (Figure 2B); however, no statistically significant difference in relative density/µm^2^ was found (Figure 2C).

The distribution of neovessels depending on the depth of the granulation tissue was also analyzed in each mouse. Their distribution patterns in the control and MG groups are shown in Figure 3A and regression lines in each group are shown in Figure 3B. In the control group with saline vehicle alone, a peak appears at approximately 60 µm depth from the wound surface. In the MG group, the density gradually elevated from deep to shallow areas, and the peak appears around the subepidermal zone.

### 3.4. Epidermal Extension and Migration

Epidermal healing was evaluated as epithelial length of different epithelial components covered on wounds, i.e., REL, CEL and TEL. Those in the MG group extended 2.8, 1.5 and 1.8 times longer than those in the control group, respectively (Table 2, Figure 4A), and REL extension was the greatest with MG. On the contrary, CEL and REL were positively correlated (Figure 4B), showing that both extended cooperatively.

CEL and REL values and granulation tissue thickness were significantly correlated (Table 1). Moreover, CEL and CD31-positive neovascularization were also positively correlated (Table 1). However, no significant correlation was found between the REL and neovascularization (Table 1).

### 3.5. Distribution of α-SMA Expressing Myofibroblasts

However, α-SMA expression was detected in arterial smooth muscle cells and peri-hair roots (Figure 5A), capillary cells in the granulation were negative. In the granulation tissue, α-SMA is primarily distributed beneath the regenerated epithelium (Figure 5A). When the border part of the epidermal regeneration and wound bed was checked using a whole mounted wound tissue from a mouse in the MG group after being treated with a tissue clearing reagent, many α-SMA-expressing cells accumulated along the front of the regenerative epidermis (Figure 5B, right), but no such cells were noted on the wound bed without epidermal healing (Figure 5B, left). The area expressing α-SMA in the granulation tissue was significantly correlated with all histological values (Table 1), especially TEL (Figure 5C). Percent area of α-SMA in the MG group tended to be larger than that in the control group (Figure 5D).

## 4. Discussion

Skin wound-healing reactions progress with fibroblast proliferation, collagen matrix formation and neovascularization in newly formed granulation tissue, on which epithelialization occurs. Therefore, a comprehensive reaction during the progress of wound-healing occurs. This becomes clear in molecular mechanisms of repair in acute and chronic wounds [22]. However, applying this knowledge to intractable human wounds is still not sufficient, because complex cellular interactions take place during the wound-healing process. In this study, we quantified and analyzed five of the tissue reactions;, i.e., granulation tissue formation, neovascularization, epidermal regeneration and migration and wound contraction. All involved different cellular mechanisms, but reciprocally interacted with each other. Our results indicate that all values showed a strong relation with each other, demonstrating that they could be synchronized, which may induce optimal outcomes of wound-healing. In the case of diabetes, such synchronization would be lost and would result in intractable wounds [23].

To date, although many clinical studies of the MG technique (Rigenera protocol) have been used to make wound-healing time more effective, research has been limited to case reports [14,15,16,17,18,24]. In most reports, the technique has been adopted as an ultimate treatment. However, more reliable controlled clinical analysis should be conducted to know the effectiveness of MG in different clinical settings. Uehara et al. [25] reported that a patient with intractable hallux ulcer with bone distraction was treated with the Rigenera technology, resulting in not only wound-healing, but also bone regeneration. In the present study, the results show that the MG technique initiates global tissue healing reactions including regenerations at the wounded site.

This study was also conducted to explore major histologic factors that lead to wound-healing using MG technique. It has been reported that grafting cells can participate in wound-healing [15]. In our preliminary study [20], the tissue suspension solution contained all of the skin tissue components including cellular elements including fibroblasts, capillary endothelial cells and muscle cells and extracellular elements including collagens. To clarify the involvement of grafted cells in granulation tissue formation, green-fluorescence protein overexpressed mice were used as donors, however grafted cells temporarily appeared in the granulation tissue, but disappeared during wound-healing. Therefore, we concluded that grafted cells could not directly participate in granulation tissue formation, but some components in the tissue suspension solution stimulate the development of granulation tissue. The extracellular components in the solution may contain cytokines [26], growth factors [6,27,28], proteinases [29] and matrix components [30], which include tumor necrotic factor, interleukins-1 and 6, transforming growth factor (TGF)-β, vascular endothelial growth factor, FGF, platelet-derived growth factor, epidermal growth factor, hepatocyte growth factor and matrix metalloproteinases, as well as collagens and collagen-derived proline-hydroxyproline peptides [31,32]. All of these factors are candidates of paracrine factors for wound-healing. Our group recently reported that the controlled delivery of various cytokines and growth factors improves tissue regeneration and wound-healing [33]. A controlled release system for simultaneous delivery of three human perivascular stem cell-derived factors for tissue repair and regeneration [34].

In this study, the skin around the wounds was initially fixed by splint, by which wound contraction could be inhibited. Our method is able to evaluate precise epithelial healing based on CEL and REL [21]. The epidermal regeneration on the granulation tissue is an ultimate target for wound-healing. To initiate and sustain the regeneration, neovascularization in the granulation tissue acts as an underlying substruction. Neovascularization occurred predominantly in the MG group compared to the control group, which was accompanied with granulation tissue development. Their distributions were also quite different:, i.e., a halfway collapsed pattern appeared in the control group, whereas an uphill pattern reaching the subepidermal zone appeared in the MG group. Functional distribution could represent the latter pattern because of the maintenance of blood supply for regenerative epidermis migrating on the granulation tissue.

Myofibroblast, a well-known phenotype of fibroblasts, carries α-SMA fibers, by which wounds can be contracted [27]. Cells are differentiated from resident tissue fibroblasts and/or blood-borne fibrocytes, and can be stimulated by various inflammatory factors, such as chemokines [35] and growth factors, especially TGF-β [27,36]. In our study, they were located in the upper area of granulation tissue, but not in the areas of contracted epidermis. The results show that myofibroblasts are distributed in a circular form in the superficial portion of granulation tissue, by which the wound can be contracted. Thus, it is reasonable to conclude that CEL and REL showed a significant correlation (Figure 4B). Recently, Balli et al. [22] showed that the MG tissue solution without cells activated the extracellular signal-regulated kinase to induce gene transductions of matrix metalloproteinases and cell migration in vitro. More recently, they found that the mechanism involved phosphorylation of c-Jun and Fos-related antigen-1 [37]. Such intracellular molecular mechanisms could also take place in epidermal healing found in our in vivo study.

## 5. Conclusions

Based on the results of this study, we concluded that the MG technique activates tissue healing potentials in wounded tissues. Different tissue reactions that appeared after wounding may be synchronized by this treatment, leading to improving wound-healing. This simple method using autologous healthy tissues can be utilized even at bedside without any special equipment for grafting other than using the rotating blade and performing the Rigenera technology. Furthermore, because this technique effectively initiates granulation tissue formation, it may be a useful pretreatment for the grafting of autologous akin, cultured epidermal cells and fat tissue-derived stem cells on intractable tissues or less granulated tissues.

## Figures and Tables

**Figure 1 medicina-56-00321-f001:**
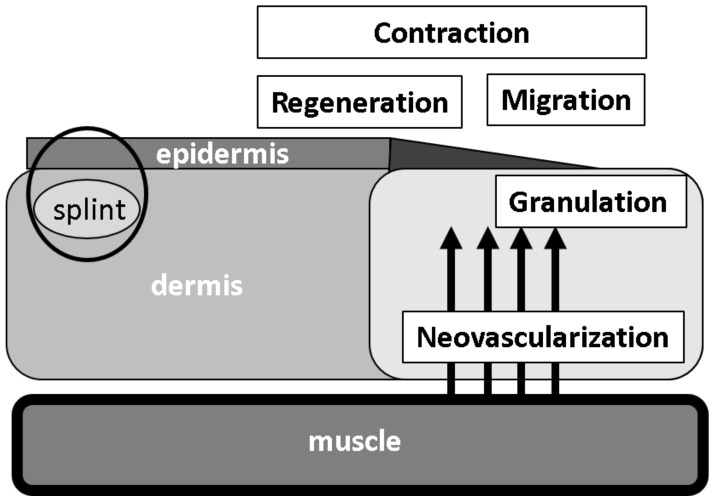
Major histological alterations for the progression of wound-healing. Granulation tissue formation, neovascularization, epithelial regeneration as well as keratinocyte migration and wound contraction are the determinants of important tissue reactions after wound injury.

**Figure 2 medicina-56-00321-f002:**
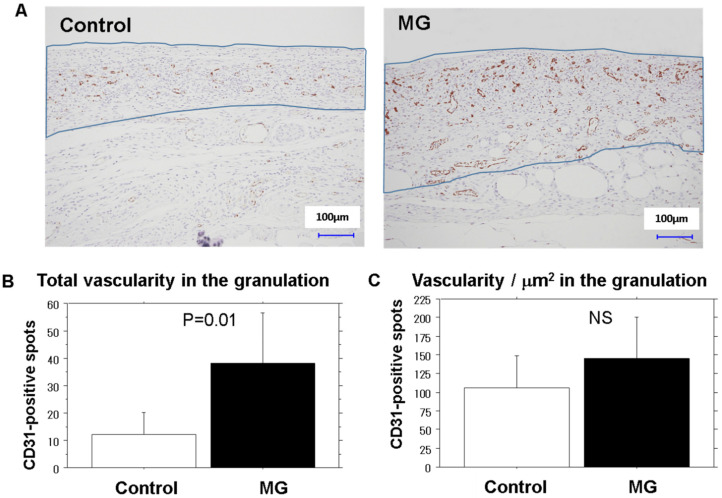
Neovascularization in the granulation tissue. Quantification of neovascularization was performed using a CD31 antibody. Detailed description of the methods is presented in the Materials and Methods. (**A**) Representative images from the control and MG (micrograft) groups. Granulation tissues are bounded by blue lines; (**B**) The number of CD31 positive cells. MG treatment increased neovascularization in the granulation more than three times compared to the control group; (**C**) Relative density of vascularity. Mean ± SD; NS: not significant.

**Figure 3 medicina-56-00321-f003:**
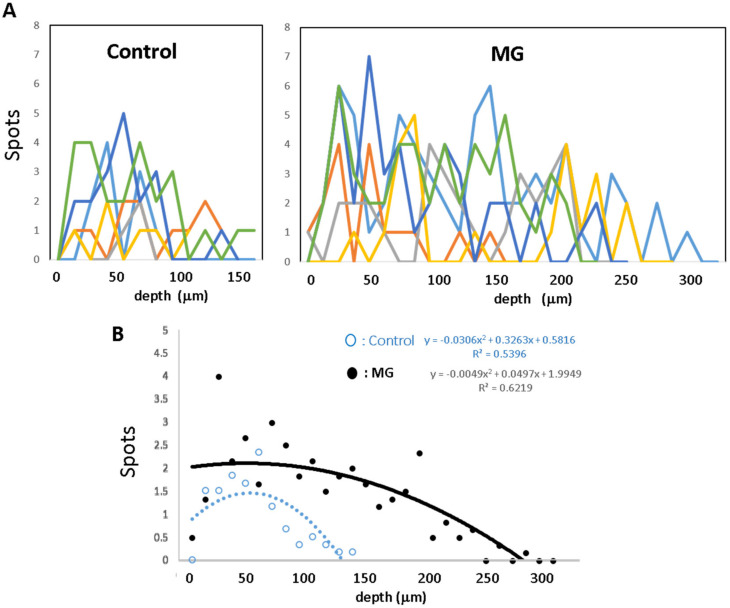
Distribution of neovessels depending on the depth of the granulation tissue. CD31-positive neovessels in the granulation tissue were plotted in each mouse, according to the methods described in the Materials and Methods. The lines in different color show different animals. (**A**) Distribution of neovessels in the control and MG groups; (**B**) regression curves of the control group and MG group. Neovessels increased in accordance with shallowing and reached the subepidermal area in the MG group but did not reach in the control group.

**Figure 4 medicina-56-00321-f004:**
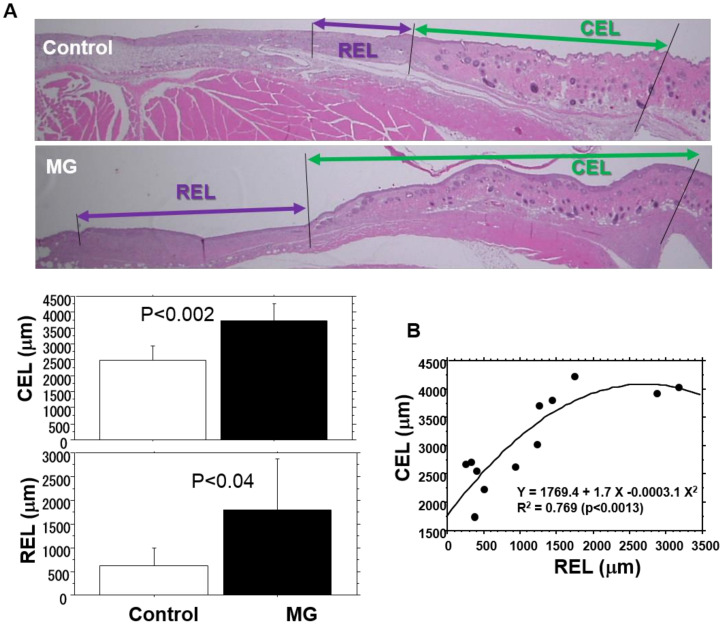
Epithelial healing on the wounds. Epithelial healing was measured in terms of contractive epithelial length (CEL) and regenerative epithelial length (REL), which were assessed by the methods described in the Materials and Methods. (**A**) Representative images from the control and micrograft (MG) groups and their mean values (mean ± SD); (**B**) correlation of CEL and REL.

**Figure 5 medicina-56-00321-f005:**
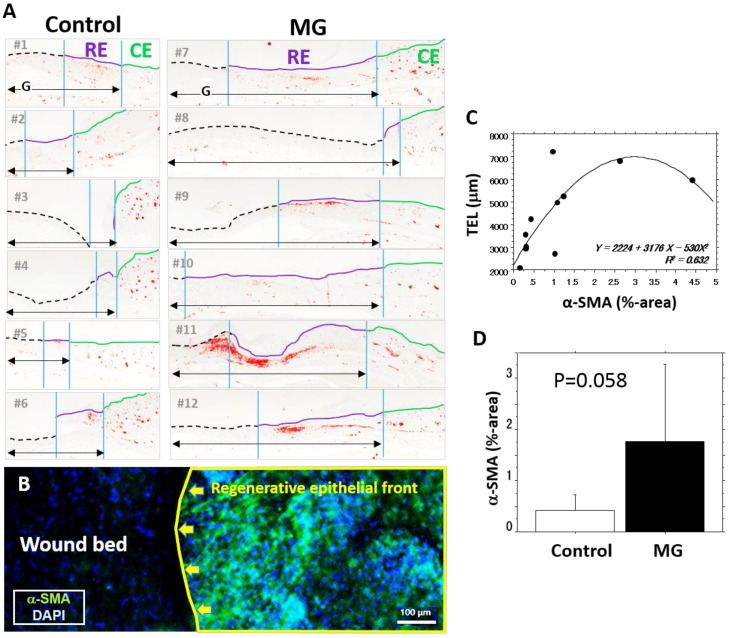
Distribution of α-SMA in the granulation tissue. Myofibroblasts expressing α-SMA play a role in wound contraction. Thus, their distribution on the granulation tissue covered by regenerative and contractive epithelium was examined. (**A**) α-SMA-positive regions in wound are shown in red. On the right side of the images, regenerative epidermis (RE) and contractive epidermis (CE) are shown by yellow and blue lines, respectively. “G” is granulation tissue; (**B**) localization of α-SMA expression in whole mounted wound tissue 12 days after MG treatment. The image was taken from above of the whole mounted wound tissue. The right side is a wound edge with regenerated epidermis (RE) and the left side is an uncovered wound bed. Nuclei were stained with 4’,6-diamidino-2-phenylindole (DAPI); (**C**) regression analysis of α-SMA (α-smooth muscle cell actin) expression (%-area in granulation) and TEL. (**D**) α-SMA expression in the control and MG groups. Mean ± SD.

**Table 1 medicina-56-00321-t001:** Histological alterations in wound-healing and their correlation coefficients.

	Wound Area (%)	Granulation Area (×10^3^ µm)	Neovessels (Points)	REL (µm)	CEL (µm)	TEL (µm)	α-SMA (Area (%))
Mean ± SD	57.0 ± 36.7	185.4 ± 92.9	25.3 ± 19.1	1213 ± 984	3098 ± 807	4311 ± 1706	1.1 ± 1.3
Wound area	–	–	–	–	–	–	–
Granulation area	−0.851 (*p* = 0.0004)	–	–	–	–	–	–
Neovessels	−0.757 (*p* = 0.0044)	0.881 (*p* = 0.0002)	–	–	–	–	–
REL	−0.932 (*p* < 0.0001)	0.790 (*p* = 0.0022)	0.643 (*p* = 0.0240)	–	–	–	–
CEL	−0.848 (*p* = 0.0005)	0.881 (*p* = 0.0002)	0.755 (*p* = 0.0045)	0.804 (*p* = 0.0016)	–	–	–
TEL	−0.911 (*p* < 0.0001)	0.881 (*p* = 0.0002)	0.762 (*p* = 0.0040)	–	–	–	–
α-SMA	−0.753 (*p* = 0.0047)	0.678 (*p* = 0.0153)	0.699 (*p* = 0.0114)	0.776 (*p* = 0.0030)	0.762 (*p* = 0.0040)	0.727 (*p* = 0.0074)	–

REL: regenerative epithelial length; CEL: contractive epithelial length; TEL: REL + CEL; α-SMA (α-smooth muscle cell actin). Values = mean ± SD.

**Table 2 medicina-56-00321-t002:** Histological alterations in the control and micrograft groups.

Histological Values	Control	MG	*p*-Value
Wound area (%)	82.2 ± 14.6	31.3 ± 35.0	0.0088
Granulation area (×10^3^ µm)	109.9 ± 34.8	260.9 ± 64.0	0.0005
Neovessels (points)	12.3 ± 7.9	38.2 ± 18.3	0.01
Epithelial healing			
REL (µm)	631 ± 371	1796 ± 1086	0.0322
CEL (µm)	2473 ± 445	3722 ± 545	0.0014
TEL (µm)	3104 ± 734	5519 ± 1538	0.006

REL: regenerative epithelial length, CEL: contractive epithelial length, TEL: REL + CEL; MG: micrograft. Values = mean ± SD.

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
