# Peer review of "Acceleration of Skin Wound-Healing Reactions by Autologous Micrograft Tissue Suspension"

_medicina, 2020, doi:10.3390/medicina56070321_

Round 1

Reviewer 1 Report

Review of Medicina MS 792428

Micrograft Tissue Suspension Accelerates Epidermal Healing by Inducing Tissue Reactions on Cutaneous Wound in Mice

Comments to authors

This is a short study, but unfortunately there are many instances of incorrect grammer, poor English, typographical errors and poor formatting of the text which require correction. The figures also need attention. Comments are listed below outlining some of these. If you choose a re-write I would recommend you enlist the services of an English speaking colleague to ensure correctness of form.                                                  

abstract

Line 21               typo     improvement                improvement the donor area

Line 22 cover the substitutes after engraftment reword

Line 22  ineluctable side outcome this term is not English re-word

Line 23  inaesthetic scars. re-word

Line 35 the granulation were re-word

Line 36  suggest reword the term “was more than 3-times greater in the MG group than in the control group”.

Line 45  “granulation formation” suggest you add the word tissue between these two words

Line 52/53   “same authors” re-phrase this

Line 53   “cot” replace with correct English

Line 56-58         “Globally, the annual medical cost for patients with chronic wounds will increase up to 3.5 billion dollars in 2021 [9]. Moreover, there is still limited accessibility to easy, reliable, and reasonable technology for poorly treated cases in the world.” –this segment needs re-phrasing

Line 61    “Granulation” suggest you use the term granulation tissue

Line 69  surely there is a better term that could be used than “hashed” macerated, minced ?

Line 74   “dehiscence” what term is this –correct it

Line 76    “Complex trauma of the limbs, producing loss of skin and soft tissue, due to domestic or working injuries.”-suggest rewording more clearly

Line 78    “introduces a fullness new concept in regenerative surgery, enduring to repair severe traumatic wounds”- re-word this segment

Line 87 the figure legend needs more information-the legend as it stands is totally inadequate

Line 99     “Our splint model was used [30].”-sentence needs expansion

Line 101   “The part of the skin was excised with scissors.” Needs incorporation with surrounding sentences.

Line 108   suspension solution should be suspension solution

Line 112      what is Rigeneracons ?

Line 113/114   “was designated as tissue solution. A part of tissue solution (100 μL) in a 96-well multi plate was used” –does not make sense-reword. There is no such thing as a tissue solution-solubilised tissue extract-tissue suspension ?

Line 117/118  “for optimal concentration of tissue solution [21], tissue conce

Line 119     “surficial cells -did you mean superficial “

Line 120/121 “mimicking a deterred clinical wound”-re-word this and also following sentence

Line 124                    explain the term WA%

Line 138                    “its” be clearer

Line 141-149           define all abbreviations at first point of use

Line 157                    “total spot numbers” did you mean pixels on the digitized image?

Line 167 “phosphate-buffered solution” did you mean phosphate buffered saline (PBS) ?

Line 170  “specimens were incubated in a tissue transparent reagent” is this a clearing agent or mountant ?

Line 182-183   “[contractive epithelial length (CEL), regenerative epithelial length (REL), and total epithelialization length (TEL)]” definitions of abbreviated terms should be used at their first point of use in the textstream.

Line 190  “Wound area (WA%)” this explanation of an abbreviated term should have been used earlier when it was first used in the text.

Line 202   The histological stain is too faint in Fig 2. The scale bars do not need to be given at an 100.0 accuracy, 100 um would have been fine.

Line 214  granulation tissue is the term that should be used. should “spots” more correctly be “histology pixel staining intensity” ?

Line 217      “halfway collapsed”-surely a better descriptive term could be used

Line 219       “Epidermal healing was evaluated as REL, CEL, and TEL.” is a rather awkward way to describe tissue changes surely a better more informative sentence could be constructed

Line 244   Fig 5A is extremely faint and very low magnification. Fig 5B is not in focus. Fig 5A needs to be described better in the legend. Can the skin surface be indicated with a fine dotted line?   These images in segment A would be considerably improved at a higher magnification.

Line 252    “Wound healing could be achieved following comprehensive reactions”-please explain better

Line 252     “Therefore, enhancing one factor alone might be insufficient”. Is not a complete meaningful sentence.

Line 267    “minced tissue solution”-inaccurate term needs replacement

Line 268      contains cellular elements and other components” ?

Paragraph Lines 269-277. English construction needs improvement.

Paragraph Lines 278-287 English needs improvement

Paragraph Lines 298-310 English needs improvement

Line 314-315   “by which any kinds of clinically intractable wounds may be adaptable.”- unclear what the authors are attempting to convey-re-word more clearly

Line 319-322  “However, despite our conclusion, these results are still limited to explain complex tissue reactions after providing MG tissue suspension to wound tissues. Therefore, we have to profoundly consider the cellular interaction between the grafted tissue and wounded tissue in our future studies.” It isunclear what this section is trying to say, this section needs to be re-written to improve its clarity.

References The reference numbers are quite modest 

Author Response

Thank you for your suggestions. Most of your suggestions have been incorporated, which are highlighted in red in the revised manuscript. 

1 English and grammatical revision: We have corrected all the words and sentences according to your advice. Please refer to the revised manuscript. 

2 Figures: especially Figs 2, 4 and 5 have been highlighted and enlarged as much as possible. The pictures on the manuscript prepared on Word look not enough resolution as they are. Thank you. 

3 References: Micrograft-related references are still quite limited. We believe that all necessary references regarding micrograft are cited. At this time, we have added additional references. 

4 The morphometrical method for neovascularization that we used was the point counting method; we checked the number of CD31-positive points on the lines that show depth of the granulation tissue. By using this approach, we can estimate neovascularization depending on the depth of the granulation tissue. Therefore, this is not a pixel-based method.

Reviewer 2 Report

micrograft tissue suspension accelerates epidermal healing by inducing tissue reactions on cutaneous w

The Authors performed an experimental study highlighting how micrograft tissue suspension accelerates epidermal healing by inducing tissue reactions on cutaneous wound in mice. The results are new, the topic interesting to a global audience.

Author Response

Thank you for useful reviewing our manuscript.

Reviewer 3 Report

Dear authors, I have completed reading your manuscript about micro graft tissue suspension. I find the idea interesting and definitely worth exploring further. My concern which you have not touched so much in your manuscript is the relevance to human wound healing and skin. As we know the human and rodent skin quite different. Not mention the underlying molecular responses. It would therefore be good if you could mention this in your introduction and your discussion as well. 

It might be worth while checking for proliferation and IL-1 activation. That would give you a better view of what's happening in the activated area.

Your English gramma and composition needs to be improved in the all parts of the manuscript. Also please label and explain the images better. That way everyone will be able to follow you better in your results and conclusions. 

Attached are some more direct comments to your manuscript. 

Author Response

Thank you for your suggestions. We have revised Figure 4 and grammatical errors, as well as added additional sentences according to your comments.

Please refer to the revised manuscript.

Round 2

Reviewer 1 Report

Review of Medicina MS 792428

Micrograft Tissue Suspension Accelerates Epidermal Healing by Inducing Tissue Reactions on Cutaneous Wound in Mice

Comments to authorsThis is a short study, but unfortunately there are many instances of incorrect grammer, poor English, typographical errors and poor formatting of the text which require correction. The figures also need attention. Comments are listed below outlining some of these. If you choose a re-write I would recommend you enlist the services of an English speaking colleague to ensure correctness of form.

abstract

Line 21               typo   improvement        improvement the donor area

Line 22               cover the substitutes after engraftment reword

Line 22               ineluctable side outcome this term is not English re-word

Line 23               inaesthetic scars. re-word

Line 35               the granulation were re-word

Line 36               suggest reword the term “was more than 3-times greater in                            the MG group  than in the control group”.

Line 45             “granulation formation” suggest you add the word tissue                                   between these two words

Line 52/53         “same authors” re-phrase this

Line 53                      “cot” replace with correct English

Line 56-58         “Globally, the annual medical cost for patients with chronic                                wounds will increase  up to 3.5 billion dollars in 2021 [9].                                Moreover, there is still limited accessibility  to easy, reliable,                            and reasonable technology for poorly treated cases in the                                 world.” –this segment needs re-phrasing

Line 61                “Granulation” suggest you use the term granulation tissue

Line 69                  surely there is a better term that could be used than                                      “hashed” macerated, minced ?

Line 74                      “dehiscence” what term is this –correct it

Line 76                      “Complex trauma of the limbs, producing loss of skin                                     and soft tissue, due to domestic or working injuries.”-                                     suggest rewording more clearly

Line 78                     “introduces a fullness new concept in regenerative                                           surgery, enduring to repair severe traumatic wounds”-                                   re-word this segment

Line 87                      the figure legend needs more information-the legend as                                   it stands is totally inadequate

Line 99                      “Our splint model was used [30].”-sentence needs                                            expansion

Line 101                    “The part of the skin was excised with scissors.” Needs incorporation with                                                      surrounding sentences.

Line 108                    suspensionsolution should be suspension solution Line

112                                    what is Rigeneracons ?

Line 113/114          “was designated as tissue solution. A part of tissue                                         solution (100 μL) in a 96-well multi plate was used” –does                               not make sense-reword.

Line 117/118          “for optimal concentration of tissue solution [21], tissue                                  conce

Line 119                   “surficial cells -did you mean superficial “

Line 120/121          “mimicking a deterred clinical wound”-re-word this and                                     also following sentence

Line 124                    explain the term WA%

Line 138                    “its” be clearer

Line 141-149           define all abbreviations at first point of use

Line 157                    “total spot numbers” did you mean pixels on the digitized image?

Line 167                    “phosphate-buffered solution” did you mean phosphate                                    buffered saline (PBS) ?

Line 170                    “specimens were incubated in a tissue transparent reagent” is this a clearing                                                agent or mountant ?

Line 182-183           “[contractive epithelial length (CEL), regenerative                                          epithelial length (REL), and total epithelialization length                                  (TEL)]” definitions of abbreviated terms should be  used                                  at their first point of use in the textstream.

Line 190                    “Wound area (WA%)” this explanation of an abbreviated term should have                                                  been used earlier when it was first used in the text.

Line 202                    The histological stain is too faint in Fig 2. The scale bars                                  do not need to be given at  an 100.0 accuracy, 100 um                                  would have been fine.

Line 214                    granulation tissue is the term that should be used.                                          should “spots” more correctly be “histology pixel                                             staining intensity” ?

Line 217                    “halfway collapsed”-surely a better descriptive term                                          could be used

Line 219                    “Epidermal healing was evaluated as REL, CEL, and                                          TEL.” is a rather awkward way to describe tissue                                               changes surely a better more informative sentence                                      could be constructed

Line 244                    Fig 5A is extremely faint and very low magnification. Fig                                  5B is not in focus. Fig 5A needs to be described better                                    in the legend. Can the skin surface be indicated with a                                    fine dotted line? These images in segment A would be                                      considerably improved at a higher magnification.

Line 252                    “Wound healing could be achieved following                                                      comprehensive reactions”-please explain better

Line 252                    “Therefore, enhancing one factor alone might be                                              insufficient”. Is not a complete meaningful sentence.

Line 267                    “minced tissue solution”-inaccurate term needs                                                 replacement

Line 268                    contains cellular elements and other components” ?

Paragraph Lines 269-277. English construction needs improvement.

Paragraph Lines 278-287 English needs improvement

Paragraph Lines 298-310 English needs improvement

Line 314-315           “by which any kinds of clinically intractable wounds may be adaptable.”- unclear                      what the authors are attempting to convey-re-word more clearly

Line 319-322           “However, despite our conclusion, these results are still                                  limited to explain complex tissue reactions after providing                                MG tissue suspension to wound  tissues. Therefore, we                                     have to profoundly consider the cellular interaction                                         between the grafted tissue and wounded tissue in our                                     future studies.” It is unclear what this section is trying to                                 say, this section needs to be re-written to improve its                                     clarity.

References              The reference numbers are quite modest and some seem                                 quite obscure.

Author Response

We would like to thank the Reviewer for the heipful feedback.

Our comments are attched, please refer the word file.

Than you.

Reviewer 3 Report

Dear authors, Thanks for the correction and additions to your new version. I have only one minor correction. In row 347, you have spelled "akin" instead of "skin". I wish you good luck with your future research. 

Author Response

We would like to thank the Reviewer for the helpul feedback.

We have corrected the word you had shown.